# Structural and Functional Comparison of *Salmonella* Flagellar Filaments Composed of FljB and FliC

**DOI:** 10.3390/biom10020246

**Published:** 2020-02-06

**Authors:** Tomoko Yamaguchi, Shoko Toma, Naoya Terahara, Tomoko Miyata, Masamichi Ashihara, Tohru Minamino, Keiichi Namba, Takayuki Kato

**Affiliations:** 1Graduate School of Frontier Biosciences, Osaka University, 1-3 Yamadaoka, Suita, Osaka 565-0871, Japan; u523219i@fbs.osaka-u.ac.jp (T.Y.); tomax002@umn.edu (S.T.); terahara.19r@g.chuo-u.ac.jp (N.T.); miya@fbs.osaka-u.ac.jp (T.M.); masamichi.ashihara@thermofisher.com (M.A.); tohru@fbs.osaka-u.ac.jp (T.M.); 2RIKEN Center for Biosystems Dynamics Research, 1-3 Yamadaoka, Suita, Osaka 565-0871, Japan; 3Faculty of Science and Engineering, Chuo University, Tokyo 112-8551, Japan; 4RIKEN SPring-8 Center, 1-3 Yamadaoka, Suita, Osaka 565-0871, Japan; 5JEOL YOKOGUSHI Research Alliance Laboratories, Osaka University, 1-3 Yamadaoka, Suita, Osaka 565-0871, Japan

**Keywords:** bacterial flagellar motility, flagellin, *Salmonella*, FljB, FliC, electron cryomicroscopy, viscosity, infection

## Abstract

The bacterial flagellum is a motility organelle consisting of a long helical filament as a propeller and a rotary motor that drives rapid filament rotation to produce thrust. *Salmonella*
*enterica* serovar Typhimurium has two genes of flagellin, *fljB* and *fliC*, for flagellar filament formation and autonomously switches their expression at a frequency of 10^−3^–10^−4^ per cell per generation. We report here differences in their structures and motility functions under high-viscosity conditions. A *Salmonella* strain expressing FljB showed a higher motility than one expressing FliC under high viscosity. To examine the reasons for this motility difference, we carried out structural analyses of the FljB filament by electron cryomicroscopy and found that the structure was nearly identical to that of the FliC filament except for the position and orientation of the outermost domain D3 of flagellin. The density of domain D3 was much lower in FljB than FliC, suggesting that domain D3 of FljB is more flexible and mobile than that of FliC. These differences suggest that domain D3 plays an important role not only in changing antigenicity of the filament but also in optimizing motility function of the filament as a propeller under different conditions.

## 1. Introduction

*Salmonella* infection is one of the four major causes of disease involving diarrheas in the world. *Salmonella enterica* serovar Typhimurium (hereafter *Salmonella*) has a wide range of hosts, and it infects not only mouse, which is its original host, but also humans. The infection occurs mainly via oral intake, and flagellar motility plays an important role in infection. The flagella enable bacteria to move through viscous environments such as mucosa, search for the host cell surface, and adhere to the cell membrane for infection [1,2].

The bacterial flagellum consists of three main parts: the basal body, which works as a rotary motor; the filament, which functions as a screw propeller; and the hook as a universal joint connecting the filament to the motor [3]. *Salmonella* has several peritrichous flagella, and the length of the filament is 10 to 15 µm long. When the cell swims straight, the motors rotate counterclockwise (CCW), and the normally left-handed supercoiled filaments form a bundle behind the cell to produce thrust. When the motors switch their rotation to the clockwise (CW) direction, the filaments switch to a right-handed supercoil in order for the bundle to fall apart so that the cell can change its orientation by tumbling to change the direction of swimming. *Salmonella* has two flagellin genes, *fljB* and *fliC*, and their expression is autonomously regulated to produce either FljB or FliC for filament formation at a frequency of 10^−3^–10^−4^ per cell per generation. This is called phase variation [4], and it is thought to be a mechanism to enable escape from the host immune system by changing flagellar antigenicity.

It has been reported that FliC-expressing bacteria display a significant advantage for invasion of most epithelial cell lines of murine and human origin compared to FljB-expressing bacteria. Differences in the swimming behaviors near surfaces have also been observed, and FliC-expressing bacteria more frequently “stop” [5]. In order to understand the differences in their antigenicity and motility function, structural and functional characterization is necessary. However, structural information is available only for the FliC filament [6,7,8].

The three-dimensional (3D) structures of FliC and the FliC filament have been studied by X-ray crystallography and electron cryomicroscopy (cryoEM) helical image analysis, respectively [6,7,8]. *Salmonella* FliC from the strain SJW1103 consists of 494 amino acid residues, and the molecular mass is about 50 kDa. The molecule consists of four domains, D0, D1, D2, and D3, arranged from the inner core to the outer surface of the filament. Domains D0 and D1 form the inner core of the filament and are made of α-helical coiled coils. These domains play a critical role in forming the supercoiled structure of the filament as a helical propeller. In addition, a β-hairpin structure in domain D1 is considered to be important for switching the conformation of flagellin subunits between the two states to produce various types of supercoiled filaments in left- and right-handed forms for swimming and tumbling [6,8]. Domains D2 and D3 are found in the outer part of the filament structure. These two domains increase the stability of the filament structure as well as the drag force of the filament as a propeller by increasing the diameter [6,9]. The outermost domain, D3, is thought to contain epitopes for antibodies, determining the antigenicity of the flagella.

To understand the differences in the antigenicity between FljB and FliC, structural information for the FljB and FljB filaments is necessary. Functional characterization of cell motility is also necessary to investigate the potentially different physiological roles played by these two types of filament, if any.

In the present study, we therefore investigated differences in the FljB and FliC filament structures and their motility functions. A *Salmonella* strain expressing FljB showed a higher motility than the one expressing FliC under high-viscosity conditions. The structure of the FljB filament analyzed by cryoEM image analysis was nearly identical to that of the FliC filament, except for the position and orientation of the outermost domain D3. Domain D3 of FljB also showed a higher flexibility and mobility than that of FliC. These differences suggest that domain D3 plays an important role not only in changing antigenicity, but also in optimizing motility function of the filament as a propeller under different conditions. We have discussed the relationship between the structure and motility function by comparing FljB and FliC.

## 2. Materials and Methods

### 2.1. Salmonella Strains

Bacterial strains of *Salmonella enterica* serovar Typhimurium used in this study are listed in Table 1. SJW1103 lacks the *fljB* operon and so expresses only the FliC flagellin. To express FljB from the *fliC* promoter on the chromosome, the Δ*fliC*::*tetRA* allele was replaced by the *fljB* allele using the λ Red homologous recombination system [10], as described previously [11], to generate the SJW1103B strain. For cryoEM structural analyses, the strain expressing the flagellar filament of the R-type straight form (the right-handed helical symmetry) was used.

### 2.2. Swimming Motility Assay

*Salmonella* strains, SJW1103B (only expressing FljB) and SJW1103 (only expressing FliC), were pre-cultured in 5 mL of Luria–Bertani broth (LB, 1% (*w*/*v*) tryptone, 0.5% (*w/v*) yeast extract, 0.5% (*w*/*v*) NaCl) with overnight shaking at 37 °C. Bacterial growth was measured via optical density at 600 nm (OD_600_).

A 5 µL measure of the culture medium was inoculated into 5 mL of fresh LB, and it was incubated for 6 h at 37 °C with shaking. The cells were diluted in the motility buffer (10 mM potassium phosphate pH 7.0, 0.1 mM EDTA, 10 mM sodium D-lactate). The viscosity of the motility buffer was adjusted by adding Ficoll PM400 (GE healthcare, Chicago, IL, USA) to a final concentration of 5%, 10%, or 15%. Swimming motility was observed using a bright-field microscope, CX-41 (Olympus, Tokyo, Japan) with a 40× objective (PlanC N 40× NA 0.65 Olympus) and a 1.25× variable magnification lens (U-CA 1.5 × Olympus) and recorded using a high-speed camera (GEV-B0620M-TC000 IMPREX, Boca Raton, FL, USA). The swimming speed was calculated using the software Move-tr2/2D (Library Co., Tokyo, Japan).

### 2.3. Fluorescence Labeling of Flagellin Antibodies

IgG antibodies against FliC and FljB were purified from rabbit serum using a Melon^TM^ Gel IgG Spin Purification Kit (Thermo Fisher Scientific, Waltham, MA, USA). The buffer was changed to a phosphate-buffered salt solution (pH 7.4) simultaneously with concentration of IgG by using a YM-50 Centriprep centrifugal filter (Merck, Darmstadt, Germany). After adjusting the concentration of IgG to 1 mg/mL, the antibodies were labelled with an Alexa Fluor™ 594 Antibody Labeling Kit (Thermo Fisher Scientific) for the FliC antibody and with an Alexa Fluor™ 488 Antibody Labeling Kit (Thermo Fisher Scientific) for the FljB antibody.

### 2.4. Immunofluorescent Staining of the Flagellar Filament

SJW1103B and SJW1103 were cultured under the same conditions used for the swimming motility assay described above. The culture was diluted 50 to 100 times with the motility buffer. To label the flagellar filaments, the cells were first attached to the surface of a slide glass coated with polylysine by filling the culture medium under a cover slip for 5 min. The culture medium was then replaced with a 20 µL solution of fluorescently labeled antibodies at 25 µg/mL, and it was left for 5 to 10 min. The excess antibody dye was washed by gently flowing 40 µL of the motility buffer twice, and the filaments were then observed by fluorescence microscopy. These observations were performed under an IX-83 optical microscope with a UPlan-SApo 100× 1.4 NA objective lens (Olympus). The number of the flagellar filaments per cell was measured using ImageJ (National Institutes of Health, Bethesda, MD, USA).

### 2.5. Flagellar Filament Purification

SJW590 was pre-cultured in 30 mL LB with shaking overnight at 37 °C, and 15 mL of the culture was then inoculated into 1 L of fresh LB. The cells were grown until the OD_600_ value reached about 1.0. The culture was collected by centrifugation and resuspended in 10% sucrose buffer (10% (*w*/*v*) sucrose, 0.1 M Tris-HCl pH 8.0). EDTA (pH adjusted by HCl to 7.8) and lysozyme were added to final concentrations of 10 mM and 0.1 mg/mL, respectively. The suspension was stirred on ice for 1 h at 4 °C, and 0.1 M MgSO_4_ and 10% (*w*/*v*) Triton X-100 were then added to final concentrations of 10 mM and 1% (*w*/*v*), respectively. After stirring for 1 h at 4 °C, 6 mL of 0.1 M EDTA (pH adjusted by NaOH to 11.0) was added. The solution was centrifuged at 15,000× *g*, and the supernatant was collected. The pH was adjusted to 10.9 with 5 N NaOH, and the sample solution was re-centrifuged at 15,000× *g* to remove undissolved membrane fractions. The supernatant was centrifuged at 67,000× *g* to collect the flagellar filament with the hook basal body attached (the filament) as a pellet. The sample was resuspended in 1 mL of Buffer C (10 mM Tris-HCl pH7.8, 5 mM EDTA, 1% (*w*/*v*) Triton X-100) and was centrifuged at 7300× *g*, and the supernatant was collected. The filament was purified by a sucrose density-gradient centrifugation method with a gradient of sucrose from 20% to 50%. The fraction containing the filament was collected and checked by SDS-PAGE. The fraction was 2-fold diluted by Buffer A (20 mM Tris-HCl pH7.8, 150 mM NaCl, 1 mM MgCl_2_), and the sucrose was removed by centrifugation of the filament at 104,300× *g*. Finally, the filament was resuspended with 10–50 µL of Buffer A, and the sample solution was stored at 4 °C.

### 2.6. Negative Staining

A 5 µL aliquot of the sample solution was mixed with 2% PTA on Pala film and was placed on a thin carbon-coated, glow-discharged copper grid. The extra solution was removed from the grid by blotting, and the grid was dried for 1 h at room temperature. The sample was checked using a transmission electron microscope, JEM-1011 (JEOL, Akishima, Japan) with an accelerating voltage of 100 kV.

### 2.7. Electron Cryomicroscopy and Image Processing

A 3 µL aliquot of the sample solution was applied to a Quantifoil holey carbon grid R1.2/1.3 Mo 200 mesh (Quantifoil Micro Tools GmbH, Großlöbichau, Germany) with pretreatment of both sides of the grid by glow discharge. The grids were blotted and plunged into liquid ethane at the temperature of liquid nitrogen for rapid freezing [9,13] with Vitrobot Mark IV (Thermo Fisher Scientific). Grids were then transferred into electron cryomicroscopes with a cryostage cooled by liquid nitrogen. The frozen hydrated specimens of the FljB filament on the grid were observed using a Titan Krios electron cryomicroscope (Thermo Fisher Scientific) operated at an accelerating voltage of 300 kV. Dose-fractionated movie images were recorded using a direct electron detector camera, Falcon П (Thermo Fisher Scientific) and were automatically collected using EPU software (Thermo Fisher Scientific). Using a minimum dose system, all movies were taken with a total exposure of 2 s, an electron dose of 10.3 electron/Å^2^ per frame, a defocus range of 0.2 to 1.9 µm, and a nominal magnification of 75,000×, corresponding to an image pixel size of 1.06 Å. All seven frames of the movie image were recorded with a frame rate of 0.1 s/frame, and the middle five frames from the 2nd to the 6th were used for image analysis. The defocus range was estimated by Gctf [14] after motion correction by RELION-3.0-β2 [15]. Using RELION-3.0-β2, the filament images were segmented and extracted in square boxes of 400 pixels with 90% overlap, and those segment images were aligned and analyzed. The overall resolution was estimated by Fourier Shell Correlation (FSC) = 0.143 (Appendix A). The summary of cryoEM data collection and statistics is shown in Appendix A.

The cryoEM 3D density map was deposited to the Electron Microscopy Data Bank under accession code EMD-9896, and the atomic model coordinates were deposited to the Protein Data Bank under accession code 6JY0.

## 3. Results

### 3.1. Motility Difference between Cells with the FljB and FliC Filaments

In order to investigate whether the two different types of flagellar filaments affected swimming motility, we first measured the swimming speed and the population of motile cells with the FljB and FliC filaments in motility buffer solutions of different viscosities adjusted by adding Ficoll. Optical microscopy was used for this observation. A wild-type *Salmonella* LT2 strain has two distinct flagellin genes, *fliC* and *fljB,* on the chromosome, and their expressions are switched autonomously. Therefore, we used two LT2 derivative strains, SJW1103 expressing only FliC and SJW1103B expressing only FljB, to examine structural and functional differences between these two flagellar filaments. These strains both have similar rates of growth (data not shown). The swimming speed of SJW1103 markedly decreased as the Ficoll concentration increased. When Ficoll was added to the final concentration of 10%, the average swimming speed decreased to 44% of that measured in the motility buffer without Ficoll. In contrast, the swimming speed of SJW1103B decreased only to 73% in the motility buffer with 10% Ficoll (Figure 1a). In addition, the decrease in the motile cell population of SJW1103B was much less than that of SJW1103. In the presence of 15% Ficoll in the motility buffer, 72% of SJW1103B cells were swimming while the motile fraction of SJW1103 cells was only 32% (Figure 1b). To examine the morphology, the length, and the number per cell of the flagellar filaments, the filaments were stained by immunofluorescence staining after 6 h of culture in LB and were observed by fluorescence microscopy. On average, most of the cells had three to five filaments, and their lengths were about 10 µm regardless of the strain (Figure 1c,d).

### 3.2. Structures of the FljB and FliC Filaments

We analyzed the structure of the FljB filament by cryoEM single-particle image analysis in one of the two mutant straight forms called the R-type. We used the R-type straight form to utilize the helical symmetry for image analysis as well as to compare the structure with the R-type straight filament formed by FliC. The 3D image of the FljB filament was reconstructed at 3.6 Å resolution from about 1.1 million segment images obtained from 2319 cryoEM movie images (Figure 2, Appendix A). It showed a tubular structure formed by a helical assembly of subunit proteins with an outer diameter of 230 Å and an inner diameter of 20 Å and with a similar helical parameter to the R-type straight filament formed by FliC (Table 2). The 11 protofilaments could also be recognized as in the FliC filament structure [6,7,8,16]. As revealed previously [6,8,16], the subunit consisted of four domains, D0, D1, D2 and D3, arranged from the inner to the outer part of the tube. Domains D0 and D1 formed the inner tubes while domains D2 and D3 formed the outer part, with domain D3 exposed on the surface of the filament. The sequence regions forming these four domains are indicated in Appendix A, in which the high sequence homology between FljB and FliC is also presented (FljB/FliC sequence alignment shows 76% identity and 1.4 × 10^−66^
*E*-value). The helical parameters of the FljB filament determined in this study were also identical to those of the R-type FliC filament [7], which is reasonable considering the high sequence homology between these two flagellins.

To build the atomic model of the FljB filament, a homology model was first generated from the atomic model of the FliC filament (PDB ID: 1UCU) (Figure 3a) [8] using MODELLER [17], then fitted into the 3D density map and refined using UCSF Chimera [18], Coot (Crystallographic Object-Oriented Toolkit) [19], and Phenix [20] (Figure 3b). The model of FljB (PDB ID: 6JY0) in the filament structure contained residues 1–192 and 288–505, which covered domains D0, D1, and D2 (Appendix A). The missing residues 193–287, corresponding to domain D3, were not built because the resolution of the 3D map was too low to trace the chain (Figure 3b). In addition, the density level of domain D3 was much lower than other domains in FljB (Figure 3e), indicating that domain D3 was more mobile than other domains of FljB. This indicates that the two-chain hinge connection between domains D2 and D3 was relatively flexible, making domain D3 mobile. The structures of domains D0 and D1 were nearly identical between FljB and FliC, with the root mean square deviation of Cα atoms being 2.8 Å (Figure 3c). Domain D2 is composed of two subdomains, D2a and D2b, and while the D2b domains (residues 356–412 for FljB and 345–401 for FliC) ware also nearly identical between the two, the D2a domains (residues177–189, 294–355 for FljB and 177–189 and 284–344 for FliC) showed a small difference in their conformations due to a four residue insertion in FljB (Asp 310–Gly 314) (Figure 3c, Appendix A).

The distinct difference in the two filament structures was the position and orientation of domain D3 relative to D2. When domains D0, D1, and D2 were superimposed between FljB and FliC, domain D3 of FljB was tilted from that of FliC by about 30°, making its position nearly 20 Å higher than that of FliC (Figure 3d). The two antiparallel chains connecting domains D2 and D3 were highly tilted in FljB to generate this difference (Figure 3c). Near this two-chain domain connection, FliC had close contacts between Ala 191, Val 283 and Asp 339, with a Cα distance between Ala 191 and Asp 339 of 4.9 Å, making the conformation of this two-chain domain connection stable (Figure 4a). Unlike FliC, FljB did not have these contacts, with the corresponding Cα distance between Ala 191 and Thr 350 being 18.4 Å (Figure 4b), possibly making the two-chain domain connection less stable and domain D3 oriented differently than FliC. As a result, domain D3 of FljB was more mobile than that of FliC, and the FljB and FliC filaments had different lateral intermolecular interactions between domains D2 and D3 of neighboring protofilaments.

In the FljB filament structure, we newly identified an interesting axial intermolecular interaction between the D0 domains that contributes to the filament assembly and stability. Along each of the 11 protofilaments, the N-terminal five residues of the subunit 1 above (the distal side of the filament) were extended perpendicular to the filament axis before forming the N-terminal α-helix, and formed an antiparallel β-strand with the extended C-terminal chain connecting domains D0 and D1 of the subunit 2 below (the proximal side of the filament) (Figure 5). The residue Gln 2 of subunit 1 and Tyr 469 of subunit 2 formed a hydrogen bond between the main chains (Figure 5c). This explains why the N-terminus is more important than the C-terminus for the filament stability [21].

## 4. Discussion

To investigate the potentially different physiological roles played by the two types of flagellar filaments formed by FljB and FliC, we carried out motility assays of two strains expressing either FljB or FliC and structural analysis of the FljB filament for comparison with the FliC filament.

There were no significant differences between strain SJW1103B, producing the FljB filament, and SJW1103, forming the FliC filament, either in the number of filaments per cell, their filament length, or their swimming speed in the motility buffer. However, their swimming speeds were clearly different under high-viscosity conditions, with a markedly smaller reduction in the swimming speed of SJW1103B than that of SJW1103 when the viscosity was increased by the addition of Ficoll in the motility buffer. These results indicated that the differences in the structure and dynamics of their flagellar filaments must be responsible for the difference in their motility.

We therefore examined the structure of the FljB filament by cryoEM single-particle image analysis in one of the two mutant straight forms called the R-type to utilize the helical symmetry. The overall structure was very similar to that of the FliC filament except for the position and orientation of the outermost domain, D3, exposed on the surface of the filament. Domain D3 of FljB also showed a higher flexibility and mobility than that of FliC. These differences suggest that domain D3 plays an important role not only in changing antigenicity but also in optimizing the motility function of the filament as a propeller under different conditions of solvent viscosity.

These differences in the relative position and dynamics of domain D3 were well correlated with the differences in amino acid sequence between FljB and FliC, which are found in regions Val 187–Gly 189 and Ala 284–Asn 285 and residues Asp 192 and Gln 282 of FliC, which form the two antiparallel chains connecting domains D2 and D3 (Appendix A). In FliC, the position and orientation of domain D3 was stabilized by hydrophobic interactions between Ala 191, Val 283 and Asp 339 (Figure 4a). On the other hand, the distances between corresponding residues in FljB, Ala 191, Val 292, and Thr 350, were much longer than those of FliC, making the stabilizing hydrophobic interactions impossible (Figure 4b). As a result, the D3 domain of FljB was more mobile than that of FliC.

How the differences in the structure and dynamics of the D3 domains on the surface of the FljB and FliC filaments contribute to their different motility functions as a propeller is difficult to answer without actually examining the role of the more mobile domain D3 via fluid dynamic simulations with these two filament structures. There must be some advantages to a higher mobility of surface domains of the helical propeller in generating higher thrust under high-viscosity conditions, such as in mucosa, which pathogenic bacteria have to swim through to reach host cells. The sequence of FljB may have been optimized for this purpose by evolution.

## 5. Conclusions

In this study, we revealed that the flagellar filament formed by FljB has an advantage over the FliC filament by allowing cells to have higher motility under high-viscosity conditions. Comparing the two structures, the subunit structures and intersubunit packing interactions were well conserved except for the position and mobility of domain D3 on the surface of the filament. The advantage of the FljB filament in the swimming motility under highly viscous conditions may have an important role in infection when bacteria must go through viscous layers of mucosa on the surface of intestinal cells or keep a biofilm in good condition, with the cells with the FljB filaments acting as a nutrition deliverer [22,23]. However, in order to clarify the relationship between the filament structure and swimming motility of the cell, further computational analyses are needed.

## Figures and Tables

**Figure 1 biomolecules-10-00246-f001:**
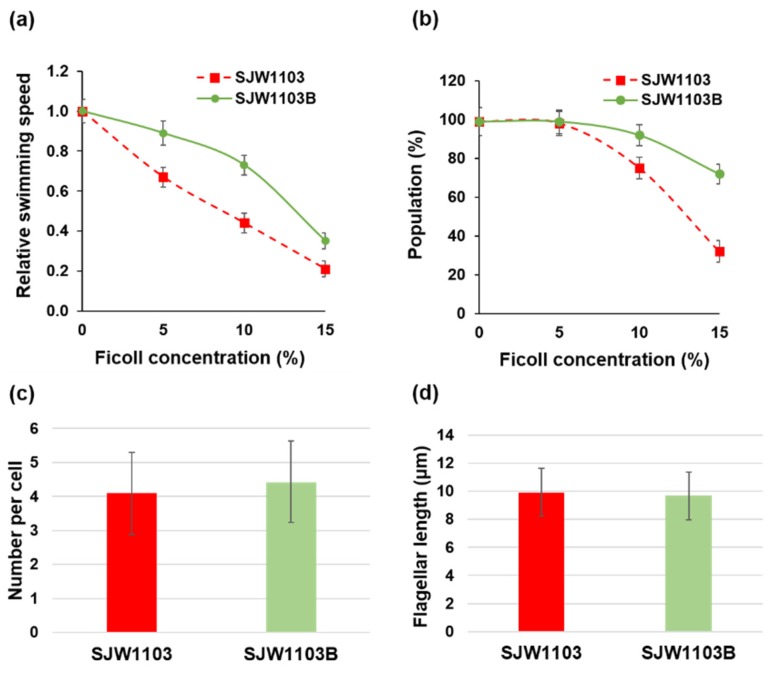
Motility and filament characteristics of *Salmonella* strains SJW1103 and SJW1103B. (**a**) Changes in the swimming speed and (**b**) changes in the swimming cell population under different viscosity conditions created by adding Ficoll to the motility buffer. (**c**) Number per cell and (**d**) length of the flagellar filaments. No significant difference was observed under either condition.

**Figure 2 biomolecules-10-00246-f002:**
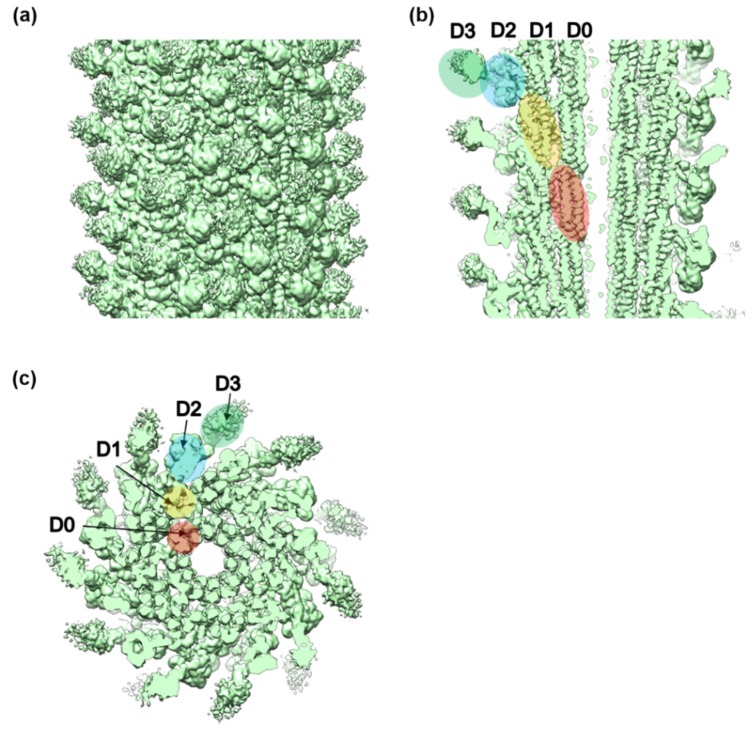
The 3D density map of the FljB filament resolved by cryoEM image analysis. (**a**) Side view of the filament surface, (**b**) longitudinal section along the filament axis, and (**c**) cross section viewed from the distal end. The four domains of FljB, D0, D1, D2, and D3, are labeled in different colors in (**b**,**c**).

**Figure 3 biomolecules-10-00246-f003:**
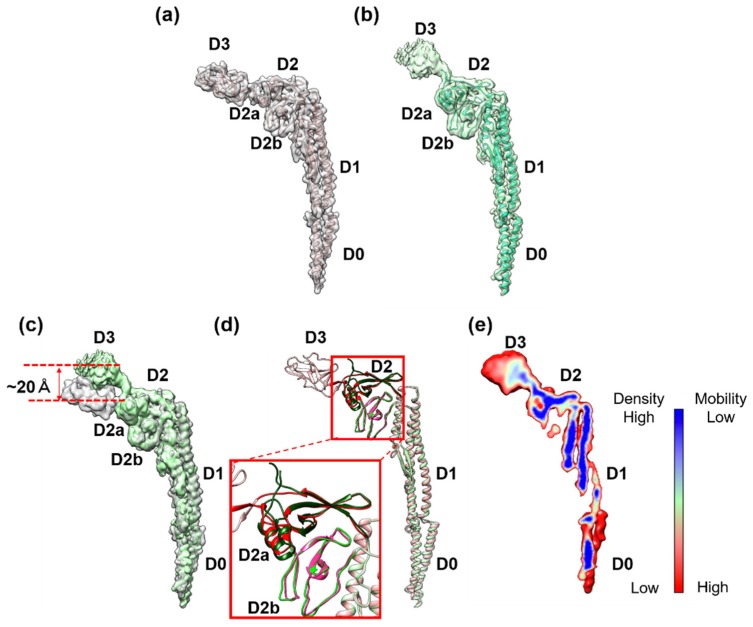
Comparison of subunit structures of the FliC and FljB filaments. The 3D map and Cα ribbon model of (**a**) FliC (PDB ID: 1UCU) and (**b**) FljB (PDB ID: 6JY0). The map of FljB was obtained by cryoEM image analysis, while that of FliC was calculated from the atomic model. The Cα ribbon model of domain D3 is not shown for FljB because the resolution of this domain was too low to trace the chain. (**c**) Superposition of the two maps with FljB in light green and FliC in gray. (**d**) The Cα ribbon model of FljB in light green is superimposed on that of FliC in light pink. The structures of FljB and FliC were nearly identical for domains D0, D1, and D2b, but the folding of part of domain D2a connecting to domain D3 was distinct between the two, which was responsible for the tilt of domain D3 of FljB, making its position higher than that of FliC by about 20 Å. (**e**) The color-coded density distribution of FljB in a longitudinal section of the molecule. The relatively low density of domain D3 indicated a high mobility of this domain.

**Figure 4 biomolecules-10-00246-f004:**
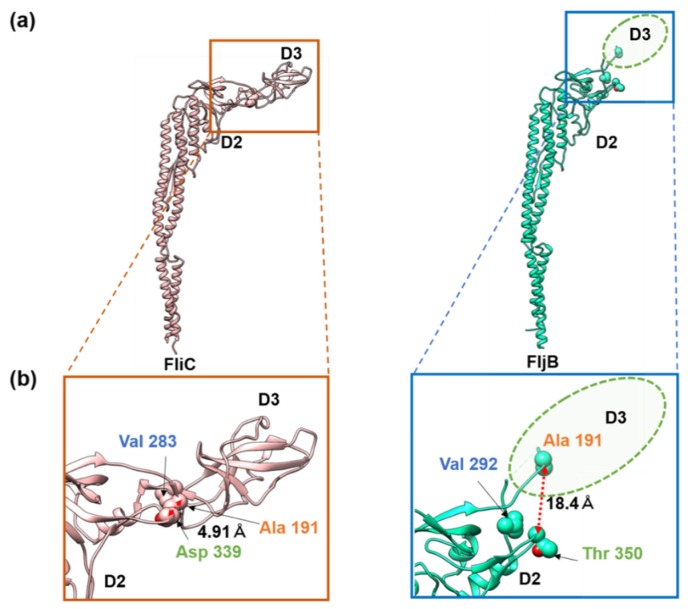
Interactions of residues stabilizing the position and orientation of domain D3 in FliC. (**a**) The Cα ribbon models of FliC (left) and FljB (right) with three residues important for stabilizing the position and orientation of domain D3 in space-filling representation. (**b**) Magnified view of the parts indicated by the boxes in (**a**). The ellipse in dashed line in the lower panel indicates the position of domain D3 of FljB. The corresponding residues of FliC and FljB are labeled in the same colors, orange (FliC Ala 191 and FljB Ala 191), blue (FliC Val 283 and FljB Val 292), and green (FliC Asp 339 and FljB Thr 350). In FliC, the three residues were closely packed each other by hydrophobic interactions, stabilizing the position and orientation of domain D3, while they were far apart in FljB, as indicated by the distances.

**Figure 5 biomolecules-10-00246-f005:**
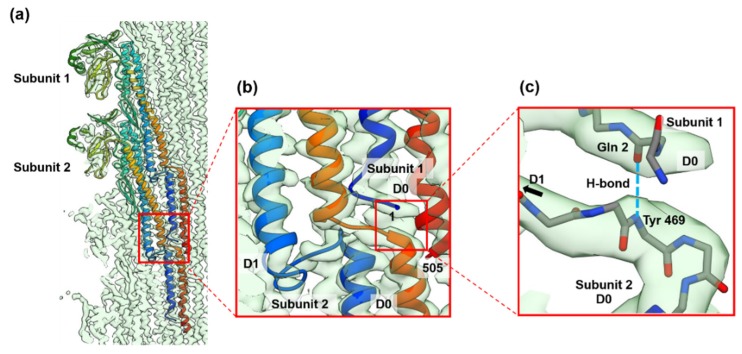
The folding and intersubunit interaction of the five N-terminal residues of FljB contributing to the assembly and stabilization of the filament structure. (**a**) The Cα ribbon models of two neighboring FljB subunits along the protofilament are shown in rainbow colors from the N-terminus in blue to the C-terminus in red. (**b**) The five N-terminal residues are extended, lying flat on the short extended chain connecting the C-terminal α-helices of domains D0 and D1 of a subunit below along the protofilament, forming an antiparallel β-strand to stabilize the axial intersubunit interactions. (**c**) The residue Gln 2 of subunit 1 and Tyr 469 of subunit 2 forms a hydrogen bond between the main chains (dashed line in right blue). In figure (**c**), only main chain backbone atoms are shown (nitrogen and oxygen atoms are colored blue and red, respectively).

**Table 1 biomolecules-10-00246-t001:** Strains and plasmids used in this study.

*Salmonella* Strains	Relevant Characteristics	Source or Reference
SJW1103	FliC wild-type Δ*hin-fljB-fljA*	Yamaguchi et al. 1984 [12]
SJW1103ΔC	SJW1103 Δ*fliC*::*tetRA*	This study
SJW1103B	FljB wild-type Δ*fliC*::*fljB*	This study
SJW590	FljB_R-type straight filament *fljB(A461V)*, Δ*fliC*	This study

**Table 2 biomolecules-10-00246-t002:** The helical parameters of the FljB and FliC R-type straight flagellar filaments.

	Rotation (Degree)	Axial Rise (Å)
FljB	65.81	4.69
FliC	65.84	4.71

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
