# Peer review of "Structural and Functional Comparison of Salmonella Flagellar Filaments Composed of FljB and FliC"

_biomolecules, 2020, doi:10.3390/biom10020246_

Round 1

Reviewer 1 Report

The intragenomic heterogeneity of flagellin, both in size and in molecular composition, is characteristic of many bacteria. However, its causes and consequences for the functioning of the cell are still imperfectly understood. Hence, the topicality of the research described and the great importance of the results obtained are obvious.

This work has a high technological and bioinformatic level. It compares for the first time the structure of the well-studied flagellin FliC from Salmonella typhimurium LT2 (PDB ID 1UCU) with its allelic variant FljB. By using strains expressing both FliC and FljB, subtle structural differences between these proteins were detected with cryo-electron microscopy. These differences were found to affect mainly the highly variable D3 domain, to have a large effect on the structure and mechanical properties of the flagellum formed from FljB, and to ensure higher cell motility in highly viscous media.

The manuscript can certainly be recommended for publication in Biomolecules. For many researchers whose interests are in the field of molecular microbiology, bioinformatics, etc., the expected publication of the atomic 3D model of the FljB flagellin of Salmonella typhimurium, submitted by the authors for deposition at the PDB database (ID 6JY0), will be interesting and useful.

One minor remark: when the authors state that there is high homology between the FljB and FliC proteins (lines 216-218 of the manuscript), I would suggest that they indicate the identity value (%) of their amino acid sequences, as well as the E-value obtained as a result of alignment (Figure S2, Supplementary Materials).

Author Response

Thank you for your positive comments on our manuscript.

One minor remark: when the authors state that there is high homology between the FljB and FliC proteins (lines 216-218 of the manuscript), I would suggest that they indicate the identity value (%) of their amino acid sequences, as well as the E-value obtained as a result of alignment (Figure S2, Supplementary Materials).

Thank you for your suggestion. We added the following sentence on lines 225 – 226 “(FljB/FliC sequence alignment shows 76% of identity and 1.4 e-66 E-value)”. We also added this information to the legend of Figure S2.

Reviewer 2 Report

Salmonella variably produces one of two flagellar filaments: FliC or FljB. While generally considered functionally analogous, there are subtle differences in motility phenotypes between filaments formed by either flagellin protein. Here, Yamaguchi and coworkers explore the structural basis behind some of these phenotypic differences. Notably, the authors discover that FljB filaments enable improved motility in fluid matrices with higher viscosity. Examining the structure of FljB, and comparing to the known FliC structure, the authors found that the D3 domain is likely more flexible and the interface between D0 domains in the FljB-based filaments are different from the D0 of FliC-based filaments. Collectively, these data increase understanding of the phenotypic and structural differences between FliC and FljB. However, connections between structural predictions and phenotype are not tested. Overall, the structural analysis and conclusions drawn were very solid and provide several directions for future studies.

Major concerns:

The study does a very nice job of demonstrating a phenotypic difference between FljB-based and FliC-based flagellar filaments. Likewise, the structural insights provide some compelling hypotheses about these differences. However, it would have been great to see one or two of these hypotheses tested. For example, the D3 domain may be the structural basis for differences in motility in high viscosity matrices. Swapping the D3 domains between FliC/FljB would be an interesting direct test of this structural prediction.

Minor concerns:

For readers unfamiliar with flagellar structural biology, explaining why FljB structures were resolved with mutants with locked R-type flagella may be helpful.

Line 60: “compare” should read “compared”.

Line 188: “These strains both showed the same growth curve (data not shown).” Perhaps write as “These strains both have similar rates and yields (data not shown).”

Author Response

Thank you for your positive comments.

Major concerns:

The study does a very nice job of demonstrating a phenotypic difference between FljB-based and FliC-based flagellar filaments. Likewise, the structural insights provide some compelling hypotheses about these differences. However, it would have been great to see one or two of these hypotheses tested. For example, the D3 domain may be the structural basis for differences in motility in high viscosity matrices. Swapping the D3 domains between FliC/FljB would be an interesting direct test of this structural prediction.

We agree that it would be interesting to see the effect of swapping the D3 domains between FliC and FljB. But unlike local or point mutations of proteins from which it is not always straightforward to predict their phenotypic changes, there is no room for doubt that the distinct difference in the motility functions between the FliC and FljB expressing strains we observed in this study is due to the differences in the structure and dynamics of the D3 domains in the two filament structures because the remaining part of the structure composed of the D0, D1 and D2 domains forming the core of the filament are nearly identical between the two (Figure 3). Therefore, we would like to carry out such swapping experiment in a future study.

Minor concerns:

For readers unfamiliar with flagellar structural biology, explaining why FljB structures were resolved with mutants with locked R-type flagella may be helpful.

We inserted the following sentence on lines 213 – 215 “We used the R-type straight form to utilize the helical symmetry for image analysis as well as to compare the structure with the R-type straight filament formed by FliC. ”

Line 60: “compare” should read “compared”.

Collected as suggested.

Line 188: “These strains both showed the same growth curve (data not shown).” Perhaps write as “These strains both have similar rates and yields (data not shown).”

We changed the sentence as “These strains both have similar rates of growth (data not shown) .”

Round 2

Reviewer 2 Report

The authors have addressed my prior concerns.